# Examination of Low-Cyclic Fatigue Tests and Poisson’s Ratio Depending on the Different Infill Density of Polylactide (PLA) Produced by the Fused Deposition Modeling Method

**DOI:** 10.3390/polym15071651

**Published:** 2023-03-26

**Authors:** Anna Gaweł, Stanisław Kuciel, Aneta Liber-Kneć, Dariusz Mierzwiński

**Affiliations:** 1Faculty of Mechanical Engineering, Tadeusz Kosciuszko Cracow University of Technology, Al. Jana Pawła II 37, 31-864 Cracow, Poland; 2Faculty of Materials Engineering and Physics, Tadeusz Kosciuszko Cracow University of Technology, Al. Jana Pawła II 37, 31-864 Cracow, Poland

**Keywords:** 3D printing, polylactide, PLA, FDM method, different infill density, hysteresis loops, low-cyclic fatigue tests, dissipation of mechanical energy, Poisson’s ratio, SEM images, DSC

## Abstract

This article examines the impact of fatigue cycles on polylactide samples produced by 3D printing using the FDM method. Samples were printed in three infill degree variants: 50%, 75% and 100%. To compere the influence of infill degree on PLA properties, several tests, including the uniaxial tensile test, the low-cycle fatigue test, differential scanning calorimetry (DSC) and scanning electron microscopy (SEM), were conducted. Poisson’s ratio has also been studied. Single hysteresis loops were summed to obtain the entire low-fatigue cycle. The infill of density influenced all compared mechanical parameters. The decrease in infill degree caused the reduction of Young’s modulus and shear modulus. For a 100% degree of sample infill, a higher number of transferred load cycles were observed compared to PLA with 75% and 50% of infill. Additionally, the value of the transferred cyclic load before fatigue failure and the dissipation of mechanical energy was the highest for 100% of infill. It is also worth noting that fatigue tests can positively affect the appearance of the PLA structure. Obviously, it depends on the number of load cycles and the infill density. It causes that if the goal is to transfer as much load as possible over a long period of time, the maximum filling of the printed element should be used.

## 1. Introduction

Three-dimensional printing technology is based on the production of elements layer by layer. This technology includes additive manufacturing (AM), rapid prototyping (RP) and solid-freeform (SFF) [1]. In order to print objects, a computer model of the object must be made in the CAD system, and then the created file is generated in the Surface Tessellation Language format. The program will make an appropriate cut into layers [2]. After selecting sufficient printing parameters, the element design made in this way can be sent to the printing machine [3].

The properties of polymeric materials produced by the fused deposition modeling method depend largely on the printing parameters [4]. In order to obtain the maximum values of strength properties, attention should be paid to the appropriate selection of the temperature value of the nozzle. Additionally, the kind of 3D printing table depends on the properties of printed material [5]. It is extremely important to determine the height of the layer, because it has a direct impact on obtaining sufficiently good strength properties [6]. It should also be noted that the possibility of using a heating chamber has a huge impact on the correct 3D printing process [7], as it guarantees the better adhesion of the printed model. Kiński et al. showed that the diameter of the nozzle has a significant impact on the tensile strength [8].

A very important aspect when designing models for 3D printing is also to select the appropriate shape of the infill density. Moradi et al., showed that a triangular infill-pattern has a maximum value of tensile strength oscillating around 15.4 MPa and a similar Young’s modulus value [9]. Rahmatabadi et al., showed that raster angle and printing velocity also influenced mechanical properties. The smallest impacts were shown to be made layer thickness and nozzle diameter [10]. Moradi et al., proved that using the same printing speed, the thickness of the layer can be reduced. This is due to an increase in strength properties and a decrease in elongation. When increasing the contour layers from two to six, it was noticed that the maximum breaking force increased by 42%. The performed analysis was also confirmed by an ANOVA analysis of variance [11].

Poisson’s ratio, similar to Young’s modulus, is the basic factor informing the mechanical properties of the materials. This coefficient is defined as the ratio of the lateral contraction ε to the elongation ε‖ and is determined by the formula: μ = −ε/ε‖. It linearly describes the elastic isotropic materials that have infinitely small deformation gradient uniaxial extensions [12]. Fereira et al. showed that polylactide with carbon fiber has a better Poisson’s coefficient than pure PLA [13]. Grasso et al. demonstrated that the Poisson’s ratio of pure PLA filaments equals 0.35 [14].

The increased mobility of the chains favors the formation of plastic deformation. The occurrence of a load causes an enhanced state of the main-chain segmental mobility of the main chain and causes plastic flow [15]. In the case of polymers, there are three stages of fatigue mechanisms. The first is plasticity-controlled failure, defined as the viscoelastic state, which arises as a result of the slow and stable propagation of individual cracks [16]. These gradually increase their size in places of stress concentration until they reach a critical value [17].

The second is slow crack growth (SCG), recognized as the slow destruction of the material. It enables planning, in terms of how a given material will decompose, and better developing the structure [18]. The cleanliness of the tested material has a great influence on the SCG. Gmbh et al., showed that the small impurities of polyethylene (PE) and polypropylene (PP) have a large impact on the fracture toughness [19]. 

The last aspect is molecular degradation [19,20]. It is associated with the breakdown of chains with the help of microorganisms [21]. This phenomenon occurs during hydrolytic degradation or the composting of polymers [22]. With time, under the influence of external factors, weight loss occurs, which contributes to the formation of microcracks and the slow degradation of the material [23].

As was showed in previous work [6], the infill density directly affects the mechanical properties. Rahmatabadi et al. also showed this dependence. The strength value of PLA90 has, respectively, 1.69 and 2.36 higher cracking resistance in relation to PLA70 and PLA50. As the content of polylactide increases, the connections between the layers and their density improve [23]. Empty spaces and the lack of complete integration of rasters between and within the layers is the reason for the deterioration of properties [24].

The aim of this work was to examine the influence of the infill density (50, 75 and 100%) of polylactide (PLA) produced by the fused deposition modeling process on mechanical properties under static and dynamic load. Basic material constants, such as Young’s modulus, Poisson’s ratio, bulk modulus and shear modulus, were calculated for three degrees of PLA infill. Additionally, the fatigue properties of the printed samples were determined. To characterize the microstructure of a filament, scanning electron microscopic analysis was conducted for samples after static tensile tests, low cycles fatigue tests and cryogenic fractures.

## 2. Materials and Methods 

### 2.1. Materials

In this work, samples with various degrees of infill of polylactide were investigated. Samples were made as 3 variants with 50%, 75% and 100% of infill density. Filaments were manufactured with a diameter of 1.75 mm by Spectrum Company (Warszawa, Poland). The samples were made on 3D printer Zortrax M200Plus 1.75 mm through the FDM method with a nozzle diameter of 0.4 mm. A heating chamber was used on the 3D printer. This type of solution is often used in the literature for eliminating material shrinkage and improving print quality and the adhesion of polylactide. PLA was printed in a typical shape with the standard samples for strength tests, in accordance with the standard EN ISO 3167.

### 2.2. Parameters of 3D Printing 

In order to obtain high strength properties, the samples were printed with a layer height of 0.14 mm. Fan speed was 20% and the retraction distance was 0.8 mm. According to the properties of polylactide based on DSC research, the temperature of extrusion was 190 °C [6]. To ensure the proper adhesion of the material the applied heating of the table amounted to 50 °C. Extrusion speed was 45 mm/s. The Zortrax printer had a nozzle diameter of 0.4 mm. All samples were made in accordance with the described parameters, differing in the infill density. 

### 2.3. Method of Testing

In this study, samples were produced from polylactide (PLA) filament spectrum at the Cracow University of Technology (Cracow, Poland). Static tensile tests (PN-EN ISO 527-1:20100) with an extensometer were conducted using an MTS Criterion Model 43 machine (MTS System Corp., Eden Prairie, MN, USA) at speeds of 2 mm/min. Differential scanning calorimetry (DSC) was performed on a NETZSCH DSC 3500 Sirius (Selb, Germany) apparatus. The specimens for the DSC tests were prepared in the same way as the samples for mechanical characterization. All the samples, weighing about 10 mg each, were cut from the oriogenic specimens. The analyses were performed under nitrogen flow (20 mL/min) from 20 °C to 200 °C. The rate of heating and cooling was 10 K/min. Based on the data determined during the DSC tests, the degree of crystallinity was calculated from the formula below [16].
(1)$$K=\frac{{∆h}_{m}-∆h_{c}}{∆h_{lit}}\times 100\%$$
K—degree of crystallinity;$∆h_{m}$—melting enthalpy;${∆h}_{c}$—post-crystalline enthalpy;$∆h_{lit}$—melting enthalpy of crystalline polymer. For 100% pure polylactide, this is equal to 110 J/g.


Low-cyclic fatigue tests were performed under stress-controlled tension–tension conditions using the hydraulic tensile machine Instron 8511.20 (Norwood, MA, USA) at a 5 Hz cyclic frequency with a sinusoidal waveform at ambient temperatures. The testing program consisted of several cyclic load blocks. The maximum force in the first block was selected in proportion to the maximum static force, starting from 30% of the maximum force for each degree of sample infill. For each subsequent load block containing 5000 cycles, the maximum force value was increased by 5%. The test was carried out to determine the fatigue destruction of the sample. Mechanical hysteresis loops were recorded and, on this basis, the dissipation energy (the area of loop) was calculated for the last loop from each loading block.

Poisson’s ratio was determined during uniaxial loading conducted with an MTS Criterion Model 43 machine (MTS System Corp., Eden Prairie, MN, USA) with speeds of 2 mm/min.

## 3. Results and Discussion

### 3.1. Bulk Modulus

Bulk modulus is a material quantity that defines the ratio of the volumetric strain to the stress occurring inside the material. The Helmholtz modulus was calculated from Formula (2) [25]:(2)$$K=\frac{E}{3(1-2\nu )}[Pa]$$
E—Young’s modulus [Pa]; ν—Poisson’s ratio.


In the case of calculations made for polylactide, the volumetric elasticity coefficient is negative. It is related to the shifting of the crystal elements under the influence of pressure. Such a phenomenon can also be seen for the alteration of chemical bonds [26]. During the measurements of Poisson’s ratio, the parameters of the transverse deformation to the longitudinal deformation were measured at the uniaxial stress state. The stretched sample tends toward decohesion over time. It is directly related to the breaking of chemical bonds and the movement of atoms under the influence of the acting force [27].

A material with a 4.5 GPa modulus of elasticity loses 1% of its volume under the external pressure of 0.045 GPa (~450 bar). This means that samples made of polylactide with 100% infill will lose 1% of their volume under the influence of a pressure of 450 bar.

### 3.2. Lame’s Parameter

In order to simplify the notation of Hooke’s law, Gabriel Lamé introduced the coefficients λ (3) and μ (4) for isotropic materials. The material factor μ is equal to the Kirchoff modulus, which is usually referred to in the literature as G, hence:(3)$$\lambda =\frac{E\nu }{\left(1+\nu )(1-2\nu \right)}[Pa]$$
(4)$$\mu =G=\frac{E}{2(1+\nu )}[Pa]$$
E—Young’s modulus;ν—Poisson’s ratio.


### 3.3. Kirchhoff’s Modulus

Shear modulus (G) is defined as the ratio of shear stress to shear strain. Knowing the value of this parameter allows us to assess the material’s resistance to shear deformation. This factor defines the elastic properties of the material. Parameters describe the relationship between the shear stress τ, which arises in the material under the influence of an external load, and the related elastic deformation, which is the shear strain angle.
(5)$$G=\frac{E}{2(1+\nu )}$$
E—Young’s modulus;ν—Poisson’s ratio.


### 3.4. Poisson’s Ratio

The Poisson’s ratio for the uniaxial stress state is the ratio of the transverse to longitudinal deformation. As a dimensionless value, it defines how the material deforms [28]. Ferreira et al., showed that this coefficient equals 0.33 for pure PLA produced by fused deposition modelling [13]. This means that the results obtained in this study for samples with 100% infill density are very similar. 

The values of obtained material factors under uniaxial loading for PLA with different infill densities are shown in Table 1. The infill density influences all compared parameters. The decrease in infill degree influences the reduction of Young’s modulus and the shear modulus.

### 3.5. Fatigue Tests

The maximum force values used for the fatigue tests according to the test program are presented below.
PLA with 100 infill density—F_max_ = 1950 N;PLA with 75 infill density—F_max_ = 1850 N;PLA with 50 infill density—F_max_ = 1625 N.The mean value of the force for all samples—Fmax = 1.8 kN


Low-cyclic fatigue tests conducted for PLA samples showed differences in the ability to transfer the load depending on the infill density (Table 2). For a 100% infill density, a higher number of transferred load cycles was observed (43,406) compared to PLA with 75% and 50% of infill (respectively 31,812 and 28,325 cycles). Additionally, the value of the transferred cyclic load before fatigue failure was higher for 100% of infill. It can be seen that PLA100 has the biggest tendency toward the dissipation of mechanical energy. This results in the scenario that if the goal is to transfer as much load as possible over a long period of time, the maximum filling of the printed element should be used.

The difference between 75 and 50% infill density is insignificant. It is worth noting that this does not apply to the maximum infill of samples. It is most likely related to the presence of empty spaces between the individual layers, which is the source of decohesion. The sample is able to transfer less energy per unit area, which results in the reduced dissipation of mechanical energy.

The energy generated during the fatigue test was calculated on the basis of the trapezoidal area dependence, assuming that the area under the curve corresponds to the energy value. The exact diagram of this method can be analyzed from a prepared diagram.

The first energy was calculated as the difference of the first two values and arguments, because the loops start at a slight distance from the X-axis. The method of calculating the energy is shown below:(6)$$E_{n}=\frac{y_{n}+y_{n+1}}{2}x_{n}$$
E—energy [J];y_n_—force value [kN];x_n_—elongation value [mm];n—number of points.


In order to determine the total energy value, the values of the individual energy fields must be added together. Data from each loop of a specific cycle were combined to obtain the final value of mechanical energy dissipation.

It can be seen in Figure 1 that samples made with 50% infill density have the lowest dissipation of mechanical energy. This is due to the presence of a large number of voids inside the material, which results in a greater likelihood of notching at the point of fissure. There is a 39% decrease in the ability to dissipate mechanical energy to the environment compared to samples made with complete infill. In the case of samples with 50% infill, there is half as much material, which reduces the possibility of energy dissipation. The voids between the filament threads cause the possibility of cracks, which are visible in the SEM photos. After exceeding 0.11 [J] of energy value, the material will be destroyed.

Printing objects with 75% infill density causes a 33% decrease in the possibility of mechanical energy dissipation compared to fully filled samples. An incomplete structure within the element has an influence on worse energy dissipation. Figure 2 shows the maximum dissipation of mechanical energy, which is equal to 0.16 [J]. This is the maximum value that determines the ultimate strength of the material. After exceeding 31,000 cycles, the material will degrade. It is worth noting that samples with 75 and 50% of infill density have small differences. Due to the longer production time of elements with a larger infill, it is worth considering the advisability of producing such elements, taking into account the time-consuming process.

The samples with 100% infill density shows the highest dissipation of mechanical energy. This is due to the exclusion of voids and the risk of notching. The maximum energy, beyond which polylactide will be destroyed, is equal to 0.2 [J]. Figure 3 shows that after exceeding this limit value, it will not be possible to conduct further cycles of fatigue tests.

### 3.6. DSC

DSC tests were performed in order to determine the temperatures characteristic for polymeric materials. Solid samples of PLA after cryogenic nitrogen cracking, clean filament and samples after fatigue tests were utilized.

In Figure 4, there are DSC charts which show characteristic temperatures for the polylactide produced in the 3D printing process and subjected to fatigue testing after cryogenic cracking and pure PLA filament. It can be seen that in the case of samples subjected to cryogenic cracking, the peak characteristic of the glass transition temperature is the most pronounced and shifted towards higher temperatures. It is most likely related to the previously performed thermal treatment and the clearer cross-linking of the structure. Moreover, the samples subjected to cryogenic cracking and fatigue tests show similar characteristics of DSC curves. Second heating for samples after fatigue tests is characterized by a smaller peak area while maintaining characteristic temperatures.

Polylactide subjected to fatigue tests has a very similar DSC curve course compared to the material subjected to cryogenic cracking in liquid nitrogen (Table 3). In this case, an increase in the glass transition temperature by 3 °C is visible [27]. In addition, the values of melting enthalpy increased by 2 J/g. It is worth noting that the glass transition is similar for each sample [28].

The degree of crystallinity directly affects the properties of polymers. A higher degree of crystallization results in a more brittle material subjected to DSC testing. This process causes the partial ordering of the chains in the molecular structure. the crystallization of the nuclei cause the molecular chains to align with each other to form ordered areas called lamellae. The crystallization of the material subjected to fatigue tests increased by 2% compared to the filament made of PLA. The prolonged cyclical loading of polylactide will cause the material to become more brittle.

### 3.7. Scanning Electron Microscope

With the aim of analyzing the microstructure of the filament thread, an image analysis on the scanning electron microscope was conducted. The samples with 100, 75 and 50% infill density after the low cycle fatigue test (Figure 5) and the cryogenic fracture (Figure 6) were investigated.

Figure 5 shows the structure of the polylactide samples after fatigue tests. Material after the cyclic load characterizes the very good regularity and adhesion between the layers. This may be due to the fact that the material is subjected to constant loading and unloading by force. Ultimately, this causes the material to harden and, as a result, it homogenizes the structure. It is also worth noting that fatigue tests can positively affect the appearance of the PLA structure. Obviously, it depends on the number of load cycles and the infill density.

Figure 6 illustrates the method of joining filament threads. It can be seen that the adhesion between the single layers is good. Additionally, the filament threads shows that the surface of individual filament threads is rough during the melting process. This is due to the slow cooling of the polylactide after it exits the nozzle.

## 4. Conclusions

In this work, the influence of the infill density on the fatigue cycles and Poisson’s ratio were determined. It can be seen that the higher infill density during the 3D printing process increase the dissipation of mechanical energy. Reducing the infill density adversely affects the fatigue properties of PLA. This is due to the occurrence of discontinuities in the manufactured products and the disadvantages of the additive manufacturing process. The low infill density increases the risk of breakage due to the presence of voids between individual filament threads. This is a potential reason for decohesion at this point. Additionally, the influence of fatigue cycles on the structure of polylactide is noticeable. The structure inside the PLA is more organized, which is related to material strengthening.

Polylactide subjected to fatigue tests has a very similar DSC curve course compared to the material subjected to cryogenic cracking in liquid nitrogen. In this case, an increase in the glass transition temperature by 3 °C is visible. In addition, the melting enthalpy values increased by 2 J/g and the values of crystallinity increased by 2%. The samples with 100% infill show the highest dissipation of mechanical energy. This is due to the exclusion of voids and the risk of notching. The degree of crystallinity directly affects the properties of polymers. A higher degree of crystallization results in a more brittle material when subjected to DSC testing. This process causes a partial ordering of the chains in the molecular structure. Crystallization nuclei cause the molecular chains to align with each other to form ordered areas called lamellae.

## Figures and Tables

**Figure 1 polymers-15-01651-f001:**
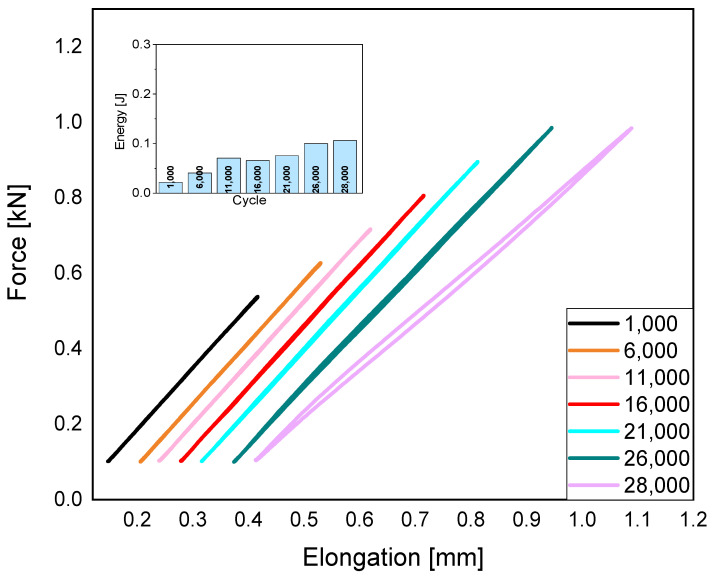
Fatigue charts, taking the dissipation of mechanical energy into consideration, for 50% infill density.

**Figure 2 polymers-15-01651-f002:**
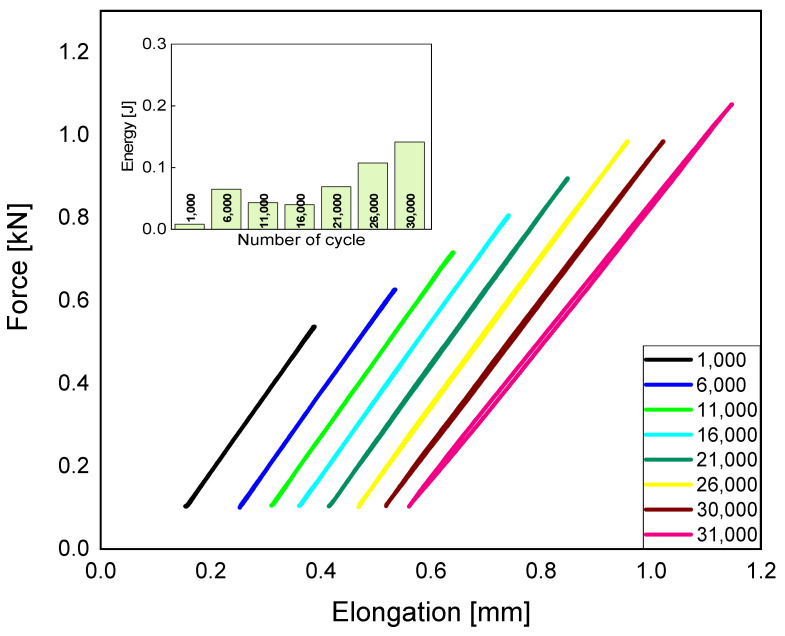
Fatigue charts, taking the dissipation of mechanical energy into consideration, for 75% infill density.

**Figure 3 polymers-15-01651-f003:**
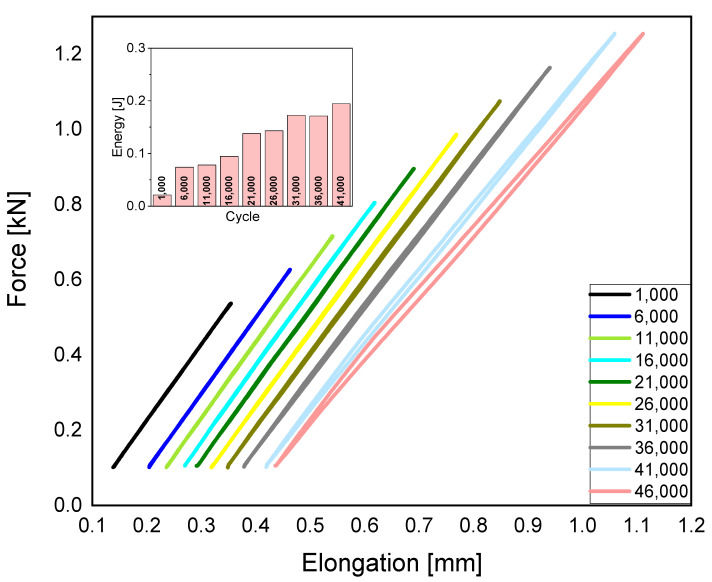
Fatigue charts, taking the dissipation of mechanical energy into consideration, for 100% density.

**Figure 4 polymers-15-01651-f004:**
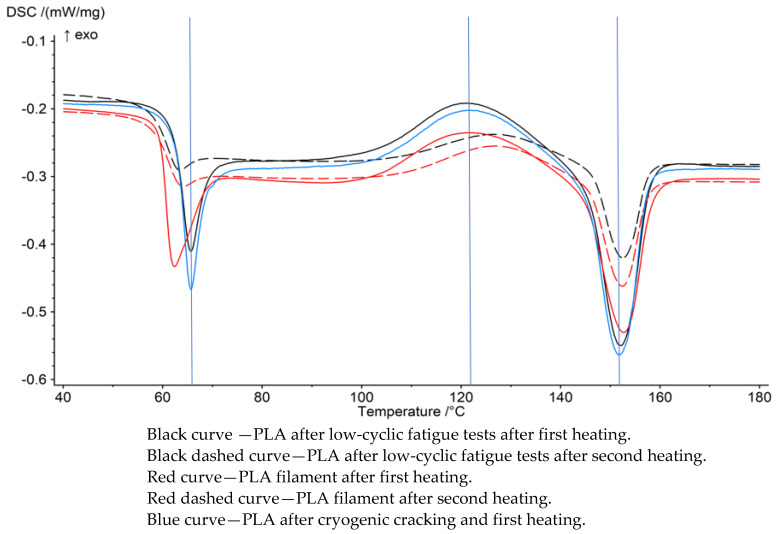
DSC curves of PLA samples produced by 3D printing.

**Figure 5 polymers-15-01651-f005:**
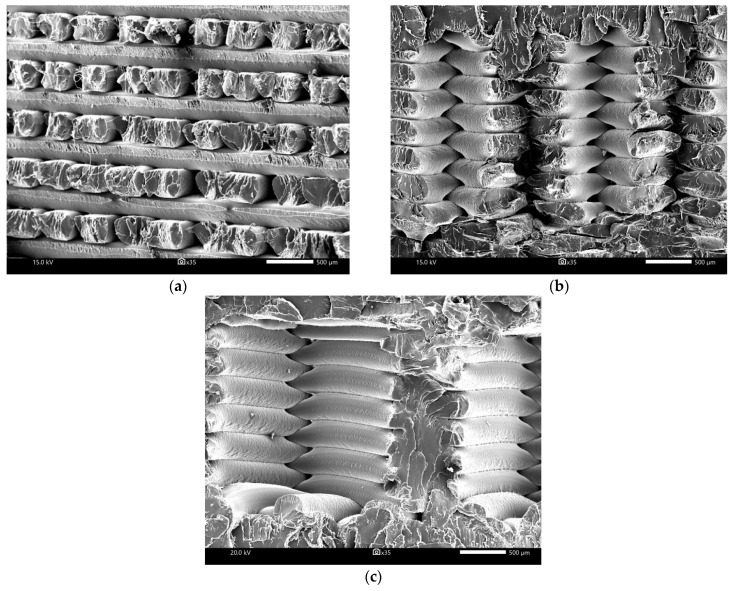
SEM images of PLA with different infill densities after fatigue tests: (**a**) 100%, (**b**) 75% and (**c**) 50%.

**Figure 6 polymers-15-01651-f006:**
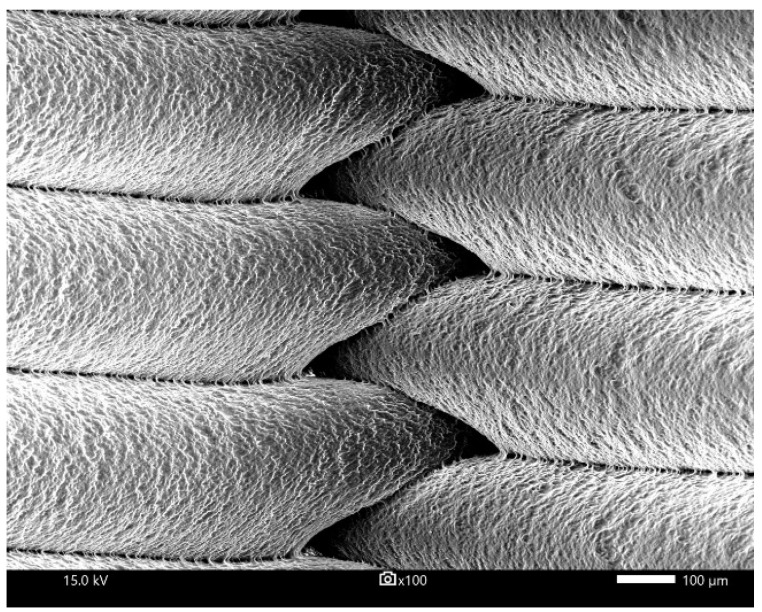
SEM images of PLA presenting adhesion between the layers after cryogenic cracking.

**Table 1 polymers-15-01651-t001:** Comparison of material factors calculated on the basis of Young’s modulus and Poisson’s ratio.

Infill Density [%]	Young’s Modulus E [MPa]	Poisson’s Ratio v	Bulk Modulus K [MPa]	Shear Modulus G [MPa]
100	2553	0.35	−4522	801
75	2188	0.38	−1137	601
50	2004	0.41	−600	478

**Table 2 polymers-15-01651-t002:** Average values of number of cycles, maximum force and dissipated energy for individual degrees of infill.

Materials	Number of Cycles	Maximum Force [kN]	Energy [J]
PLA50	28,325	0.99	0.86
PLA75	31,812	1.08	0.95
PLA100	43,406	1.26	2.19

**Table 3 polymers-15-01651-t003:** Comparison of characteristic temperatures from DSC tests for pure polylactide filamentafter fatigue tests and cryogenic cracking.

Specimen	Glass Transition [°C]	Energy [J/g]	DSC mW/mg	Cold Crystallization [°C]	Energy [J/g]	DSC mW/mg	Melting Point [°C]	Energy [J/g]	DSC mW/mg	Calculated Crystallinity [%]
PLA_100_A	65.8	−3.737	−0.4403	121.7	16.2	−0.2025	151.8	−9.978	−0.5581	16.2
PLA_100_Z	65.7	−2.91	−0.4102	121.4	16.53	−0.1915	152.1	−9.112	−0.5498	16.5
PLA_filament	62.4	−4.995	−0.4331	122.0	14.72	−0.2353	152.6	−8.272	−0.5306	14.7

PLA_100_A—samples after cryogenic cracking. PLA_100_Z—samples after fatigue tests. PLA_filament—pure PLA filament.

## Data Availability

Not applicable.

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
