# Peer review of "Examination of Low-Cyclic Fatigue Tests and Poisson’s Ratio Depending on the Different Infill Density of Polylactide (PLA) Produced by the Fused Deposition Modeling Method"

_polymers, 2023, doi:10.3390/polym15071651_

Round 1
Reviewer 1 Report
Overall, the article appears to provide a comprehensive introduction to the 3D printing technology and its applications, with a particular focus on the properties of polymeric materials produced by the Fused Deposition Modeling (FDM) method. The article also discusses the influence of the printing parameters on the mechanical properties of printed objects, including the selection of temperature value, type of printing table, layer height, and the use of a heating chamber.
The aim of the study is also well-defined, and the article provides a clear outline of the experimental setup and methodology. The use of SEM analysis to characterize the microstructure of the filament after different tests is a good approach to understanding the influence of the degree of infill on the mechanical properties of the printed objects.
Overall, the article appears to be well-written and informative. However, there are a few areas where it could be improved:
- The article could benefit from more references to support some of the claims made, especially in the introduction and the section on the properties of LA.
- The article could benefit from more detailed explanations of some of the technical terms used, especially for readers who may not be familiar with 3D printing or material science.
- Section 2 jumps from discussing the geometry of the samples to the necessity of using a heating chamber without clear transition or explanation.
- The sentence "According to the properties of polylactide, the temperature of extrusion was 190°C" could benefit from further explanation or context about the specific properties being referred to.
- The sentence "Maximum wall thickness was 3.13 mm and first layer gap was 0.31 mm" seems out of place and could be better integrated into the paragraph.
Author Response
Dear Reviewer,
Thank you very much for your review. We are delighted for your recommendation to publish our article in the Polymers Journal. As you and another Reviewer suggested, we have tried to look at the problem of more clearly present our result. Based on the bibliography suggested by the other reviewers and additionally searched for needs of re-analysis, we decided to carry out more detailed analysis of the state of knowledge and the results of works by another authors. All of changes had been marked by yellow color.
The list of corrected elements of the article:
- One part of introduction about LA-acid has been removed.
- As suggested, the results are described in more detail.
- Section 2 concerning samples geometry has been corrected.
- The sentence "According to the properties of polylactide, the extrusion temperature was 190°C" was explained in context related to specific properties.
- Phrase “Maximum wall thickness was 3.13 mm and first layer gap was 0.31 mm” has been deleted.
- The title and all the text has been changed from “degree of infill” to “infill density”.
- The abstract has been changed.
- The purpose of doing the work is mentioned in the abstract.
- As suggested the keywords were reparsed with the abstract
- We included in the manuscript following references as sugessted:
- Experimental investigation on mechanical characterization of 3D printed PLA produced by fused deposition modeling (FDM)
- Development of Pure Poly Vinyl Chloride (PVC) with Excellent 3D Printability and Macro‐and Micro‐Structural Properties
- Assessment of controllable shape transformation, potential applications, and tensile shape memory properties of 3D printed PETG
- 3D printing of PLA-TPU with different component ratios: Fracture toughness, mechanical properties, and morphology
- Statistical and experimental analysis of process parameters of 3D nylon printed parts by fused deposition modeling: response surface modeling and optimization
- Figure 1 was deleted.
- The data in the table have been calculated on the basis of mathematical formulas. Poisson's ratio was described in our article in accordance with literature references. They are consistent with the results obtained by other researchers. The static tensile test of PLA has already been described in a previous article published in the Polymers Journal. As such, we saw no need to duplicate the mechanical results again. On the basis of the forces obtained from the static tensile test, the parameters of fatigue cycles were also selected, as described in the fatigue tests section.
- All chapter numbers have been reviewed and corrected.
- Added analysis of scientists in the research section
- The conclusion was modified.
We are honored to receive favorable but constructive comments. We hope they now meet your requirements.
If you have any suggest, please do not hesitate to answer.
Yours sincerely,
Anna Gaweł,
Aneta Liber-Kneć,
Dariusz Mierzwiński,
Stanisław Kuciel.

Reviewer 2 Report
The title needs correction. Infill density is usually used instead of degree of infill.
The abstract is written very briefly and superficially and lacks quantitative results, novelty and achievements. The abstract should appeal to the reader. Novelty should be presented transparently. Research achievements should be mentioned quantitatively. The purpose of doing the work should also be mentioned. The current version presents generalities, research methods, and some qualitative results.
Be careful in choosing keywords. Most of them are not used at all in the abstract. (hysteresis loops; dissipation of mechanical energy; Poisson’s ratio; SEM images; DSC).
All introductions must be rewritten. The third paragraph is general that is not suitable for a research paper. The introduction is very incomplete and should be presented in more depth.
Use the new and relevant articles suggested in this field to strengthen the introduction. (Experimental investigation on mechanical characterization of 3D printed PLA produced by fused deposition modeling (FDM), Development of Pure Poly Vinyl Chloride (PVC) with Excellent 3D Printability and Macro‐and Micro‐Structural Properties, Assessment of controllable shape transformation, potential applications, and tensile shape memory properties of 3D printed PETG, 3D printing of PLA-TPU with different component ratios: Fracture toughness, mechanical properties, and morphology, Statistical and experimental analysis of process parameters of 3D nylon printed parts by fused deposition modeling: response surface modeling and optimization).
Figure 1 should be deleted. The text below should be modified. Referencing should be corrected. The most important part of printing is choosing the optimal printing parameters. First, the selected parameters are summarized in a table and how to choose the parameters will be explained.
How are the reproducibility of mechanical properties results checked? What is the reason for choosing equations 2 to 5? On what basis are the parameters of this equation chosen? Why are the tensile test results not provided? How is the accuracy of the data in the tables checked?
Why does fatigue have a separate section number? Correct the numbering of sections and subsections. The way of reporting the results is very unprofessional and beginner. This section needs to be rewritten.
SIM images are raw and should be used to justify results. The scale bar is not transparent. In the results section, in addition to presenting experimental data, discussion and analysis should also be added to it.
The conclusion should be modified as in the abstract section.
Author Response

(The authors gave the same response as above.)

Round 2
Reviewer 1 Report
Accept in present form
Reviewer 2 Report
Accept